

# The Friction of tilted Skates on Ice

**J. M. J. van Leeuwen**

Instituut-Lorentz, Universiteit Leiden,
Niels Bohrweg 2, 2333 CA Leiden, The Netherlands.

## Abstract

The friction felt by a speed skater is calculated as function of the velocity and tilt angle of the skate. This calculation is an extension of the more common theory of friction of upright skates. Not only in rounding a curve the skate has to be tilted, but also in straightforward skating small tilt angles occur. The tilt increases the friction substanstially and even for small tilts the increase is relevant. The increase of the friction with the velocity, which is very slow for the upright skate, becomes more pronounced for large tilts.



# 1   Introduction

Skating is an intriguing sport from the physics viewpoint, as ice seems to be the only substance that allows skating in a remarkable range of temperatures, velocities and skater weights. The physical problem is twofold: the ice should allow to push oneself forward and the friction should be low enough to glide. This is achieved by the form of the skate which has sharp edges and a thin profile, leading to a small friction in the forward direction and a large friction in the sideways direction. It is mainly the small friction in the forward direction that begs for an explication. Indeed the measured friction forces are orders of magnitude lower than the friction between steel and another solid medium. In spite of the fact that skating has been around for centuries, there is still no consensus on the friction mechanism. One school of thought [1–3,5] holds that the key to skating is the structure of the surface of ice that is "wet" within the temperature range from 0 to -30 centigrade. "Wet" means that the surface layers of ice are very mobile. The similar Arrhenius behaviour of the surface mobility and the friction supports this explanation.

The more conventional school of thought [6–9] explains the low friction by the formation of a liquid layer between skate and ice. In [7] the history of the various explanations for the possibility of skating are presented and the arguments in favour of frictional melting are summarised: skating occurs at temperatures relatively close to the melting temperature such that the friction provides sufficient heat to melt a thin though macroscopic layer of ice at the contact surface of skate an ice. The layer is also thicker than the asperities on a well polished skate. Frictional melting is a self-stabilising mechanism: if the friction becomes higher the molten layer thickens and lowers the friction, while a tendency to lower friction yields a thinner layer with higher friction. The two types of explanation, wet surface and frictional melting, are not in conflict with each other and could cooperate, in particular at the tip of the contact.

This paper deals with the same problem as discussed in [7]: the friction of a tilted skate. The reason to return to this problem is that in our opinion the used rheology needs refinement. The key quantity in deforming ice is its hardness $p_{\mathrm{h}}$, which is the limiting pressure for elastic deformations. Above $p_{\mathrm{h}}$ the ice deforms plastically. As skates leave a visible trail behind, the deformation is clearly plastically. The issue is the reaction rate of ice to an applied pressure $p$. We propose the following (Bingham) relation

$$v_{\mathrm{ice}} = \gamma(p - p_{\mathrm{h}}),\tag{1}$$

where $v_{\mathrm{ice}}$ is the (downward) velocity of the ice surface and $\gamma$ is a material constant with the dimension [m/(Pa s)]. In Eq. (1) the pressure $p$ is supposed to be larger than $p_{\mathrm{h}}$. For lower pressures the ice does not indent plastically. Eq. (1) interpolates between rheologies used in the other studies. [8] assumes that ice does not indent due to pressures no matter how high. Only melting causes a furrow, which means $\gamma = 0$. In the papers [6,7] it is assumed that the pressure stays equal to the hardness, no matter how fast $v_{\mathrm{ice}}$. This is achieved for $\gamma \to \infty$, since then the pressure stays equal to $p_{\mathrm{h}}$.

The main impact of Eq. (1) is that it leads to higher pressures on the ice, as a consequence of substantial downward velocities $v_{\mathrm{ice}}$, of the order of centimetres per second. The larger $\gamma$, the closer $p$ stays to the hardness. Higher pressures shorten the contact surface (length) and therefore lower the friction. Thus the rheology described by Eq. (1) will lead to a friction increasing with the value of $\gamma$.

Another consequence of the plastic deformation of ice in skating is that the boundary conditions for the pressure in the water layer differ from those used in [6,7,9] where the pressure is set to zero at the boundary of the water layer. We will argue that it should be equal to the hardness $p_{\mathrm{h}}$ at the transverse boundaries of the bottom water layer.

Most studies deal with the friction of an upright skate. However the upright position is

rather rare in skating. Even in straightforward skating the skate mostly has a small tilt angle. In curves the tilt angle may be very large i.e. substantial more than $45^0$. There are few measurements of the friction in real skating. The only measurements to date are of de Koning et al. [10], which indicate that the upright position gives the lowest friction, but detailed values of the friction as function of the tilt angle are lacking. The upright position has obtained the most attention as it is easier to treat because of the left-right symmetry and the way the ice is touched. In this note we extend the earlier study [9] for upright skates to tilted skates.

The tilt angle may result from two reasons: one is that of the beginner in skating, seeking stability from the large transverse friction. An experienced skater on the other hand has the skate permanently in line with the legs, such that the system of body and skate can be considered as rigid. We are interested in the latter case. Fig. 1 shows a speed skater in a curve with a rather large tilt angle. (Short trackers experience even larger tilt angles in the curve of the track, which has a shorter radius of curvature.) The body needs for stability a tilt angle parallel to the resultant of the gravitational force $Mg$ and the centrifugal force $MV^2/R_c$, where $M$ is the mass of the skater, $V$ the velocity and $R_c$ the radius of the curve. These forces act on the center of mass of the skater and must be compensated by an equal and opposite force from the ice exerting on the skate. From this equilibrium we can calculate the tilt angle $\psi$ of the skater

$$\tan \psi = \frac{V^2}{gR_c}. \tag{2}$$

Although the skate is perfectly in line with the standing leg, the tilt angle of the *skate* is *not* the same as that of the *body* (defined by the line from the skate to the center-of-mass of the skater).

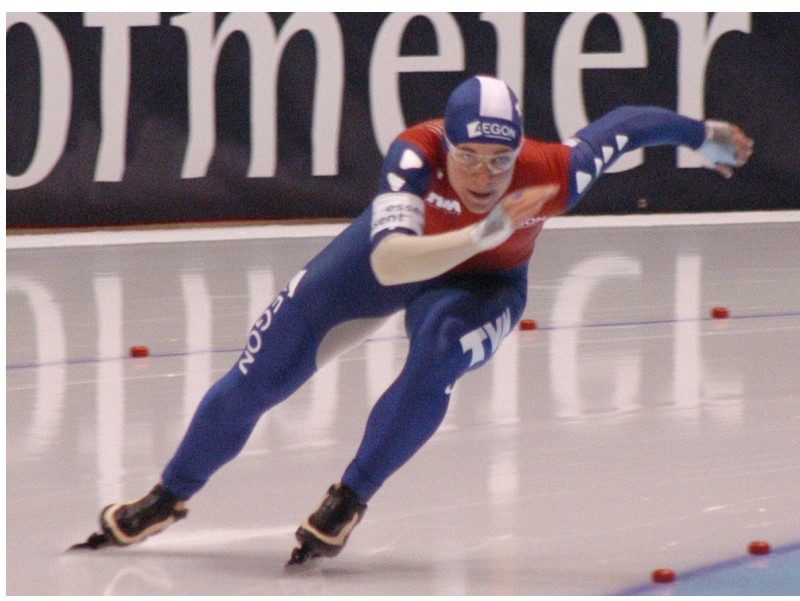

Figure 1: A speed skater rounding a curve. Photograph of the skater Paulien van Deutekom, by McSmit [CC BY-SA 3.0 Wikimedia Commons].

For the forces on the skate we need the tilt angle $\phi$ of the skate. As the Fig. 1 shows, $\psi$ will be somewhat smaller than $\phi$. We will calculate the difference. This implies that not only the basis of the skate feels a pressure, but to a lesser extend, also the side. Actually the way in which the skate is pushed against the ice is complicated, as it can be varied by the muscles in the foot of the skater, without changing the overall forces as weight and centrifugal force. The

pressure can be shifted from front to rear and from basis to side. Moreover the equilibrium to which we alluded in Eq. (2) does not need to be realised as the skater can shift weight from one foot to the other. Here we leave all these nuances aside and concentrate on the friction of a skate which makes a tilt angle $\phi$ with the normal and we do not consider force components in the forward direction other than the friction between the skate and the ice.

We study the friction of a skate which is mainly pushed in the ice in the direction of the tilt (see fig. 2). Lozowski et al. [6,7] discuss the the various ingredients in the melting mechanism. Here we restrict ourselves to the two main influences: frictional melting and squeeze flow. The others, e.g. the heat flow in the ice and skate, are of lesser importance and left out. There are many parameters, such as: temperature, weight, speed, mass and type of skates, which all can be varied, leading to myriad of cases. In order to avoid this, we focus on "standard" skating conditions: the skater's mass $M = 72$ kg, the velocity $V = 10$ m/s, the curvature of the skate blade $R = 22$ m, the width of the blade $w = 1.1$ mm and the temperature $T$ a few degrees below freezing, which we acknowledge by the choice for the hardness of ice $p_h = 10$ MPa. Unfortunately the data in the literature on the hardness of ice show a large variation [2, 12, 13]. The choice $p_h = 10$ MPa is a compromise. For $\gamma$ we take, somewhat arbitrarily, the value $\gamma p_h = 2$ mm/s in the calculations.

Our interest is the tilt angle dependence of the friction. After introducing the useful coordinate system, the pressure in the water layer is derived from the hydrodynamic equations. A hydrodynamic treatment is relevant since the water layer has a thickness of the order of a $\mu$m. Important for the solution are the boundary conditions, for which we derive expressions in a separate Section. The calculation of the thickness of the water layer proceeds along similar lines as in the case of an upright skate [9]. The water layer at the basis and the side have to be discussed separately, due to the different role and boundary conditions. The paper closes with a presentation of the results and a discussion of the main features of the solution.

## 2 The Geometry of the Indentation

An upright skate has three contact surfaces with the ice: one at the bottom of the skate and two at the sides. A tilted skate mostly has two contact surfaces: one at the bottom and one at the side. Below a small critical tilt angle $\theta_c$, one second side surface appears. We leave out here the regime $0 < \theta < \theta_c$ as it is small and near $\theta_c$ the friction is already close to that of the upright skate. A picture of a transverse section of the skate is drawn in fig. 2.

The force drives the skate into the ice mainly in the direction of the tilt. We separately treat the two different surfaces between skate and ice. The bottom surface ploughs a furrow in the ice while the skate moves parallel to the side surface. In order to describe them it is convenient to use a coordinate system $(x, \zeta, \xi)$ which is rotated around the $x$ axis over the tilt angle $\phi$, see Fig. 2. The $x$ axis runs along the line where the side of the skate meets the ice. The coordinate $\zeta$ runs along the side of the skate upwardly. The $\xi$ coordinate runs along the bottom of the skate. So for the side water layer the $\zeta$ measures the width of the layer and the $\xi$ the thickness. For the bottom layer the $\zeta$ measures the thickness and the $\xi$ the width.

The length over which the skate makes contact with the ice the along $x$ axis is the contact length $l$. $l$ is of the order of centimetres and thus much smaller than the radius $R$ of the curvature of the skate.

The side contact surface of the skate has a simple form as it is flat and bounded by a straight line and a circle. The straight line is the intersection of the surface of the ice and the side of the blade The circle is part of the edge of the skate and thus has the radius of curvature $R$. In Fig. 3 (a) we have drawn the side surface. In formula the side surface is in the $(x, \zeta)$ and

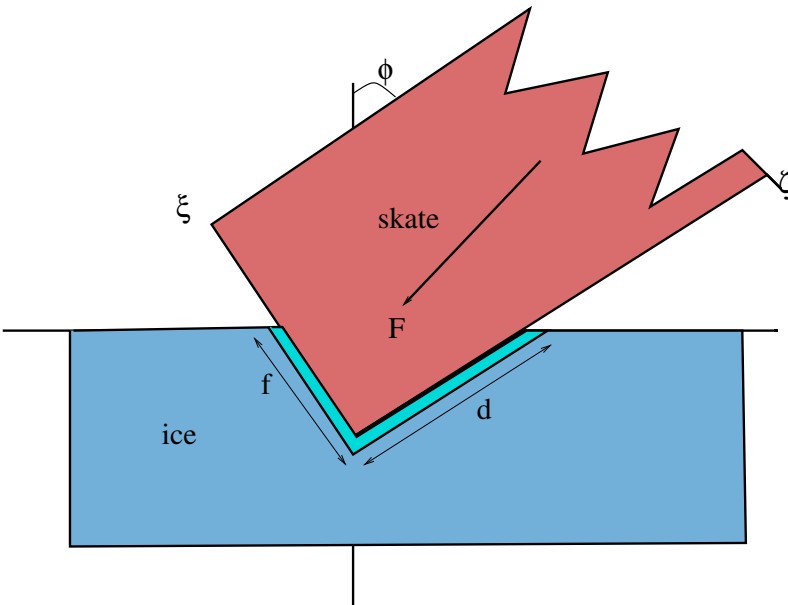

Figure 2: Tilted Skate indenting the Ice

bounded by the line $\zeta = 0$ and the circle

$$\zeta_e(x) = d\left(\frac{x^2}{l^2} - 1\right) \qquad \text{or} \qquad x_e(\zeta) = l\sqrt{1 + \zeta/d}. \tag{3}$$

Note that $\zeta_e$ is negative. Here $d$ is the deepest intrusion distance in the middle of the skate and relates to the contact length $l$ as (approximating the circle by a parabola)

$$d = \frac{l^2}{2R}. \tag{4}$$

The bottom contact surface is more complicated as it is part of the cylindrical bottom surface of the blade. As this cylinder has a very large radius the bottom surface is nearly flat. On one side it is bounded by the edge of the skate and on the other side by the intersection of the ice surface and the bottom cylinder. In the $(x, \xi)$ plane it is bounded by the line $\xi = 0$ and the circle

$$\xi_e(x) = f\left(1 - \frac{x^2}{l^2}\right) \qquad \text{or} \qquad x_e(\xi) = l\sqrt{1 - \xi/f}, \tag{5}$$

with $f$ the distance of largest width

$$f = \frac{d}{\tan(\phi)}. \tag{6}$$

When $f$ equals the width $w$ of the blade we reach the critical tilt $\theta_c$. In Fig. 3 (b) we have drawn the bottom contact surface. We have given the coordinates of the points on the edge the subscript $e$. We represent them either by the pair $(\zeta_e(x), \xi_e(x))$ or reversely as $x_e(\zeta)$ or $x_e(\xi)$.

In Fig. 3 we have given different colours to the foremost half of the bottom surface and the other half, as in the former the skate is separated from the ice by a melted layer and the latter also a layer of air is in between. In the side surface there is all over a layer of water in between.

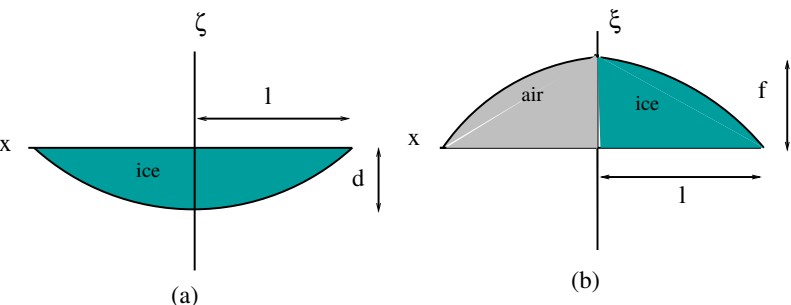

Figure 3: Shape of side (a) and basis (b) surface.

# 3 Boundary Conditions

There is ample evidence that a skate deforms the ice plastically. The skate leaves a visible trail behind in virgin ice and a skating rink has to be mopped up regularly in order to improve the skating conditions. Recently Th. Boudewijn [11] has carefully measured the indentation that a skate (in the upright position) leaves behind after it has been pushed into the ice. He finds a trough with sharp walls at the position of the edges of the skate. So the indentation is not elastic but plastic. The skate ploughs a furrow in the ice, which is as deep as the skate penetrates. Thus we ignore possible elastic deformations.

The flow in the water layers is dominantly that of a Couette flow in the forward ($x$) direction, which is sheared on top by a skate with velocity $V$ in the $x$ direction and sticks to the ice at the bottom. The counter forces of the ice are mediated to the skate by water layers. Therefore we have to know the pressure distribution in the water layers. In Appendix A we summarise the standard Couette theory for the velocity field $\mathbf{v}$ and the pressure $p$ in a layer of water of thickness $h$. There is a force on the layer in the downward direction and due to the resulting pressure, the water also flows in the transverse $y$ direction, with a much smaller velocity $v_y$. The water layer is pushed down at the top with a velocity $v_{\text{sk}}$ and the ice yields with a rate $v_{\text{ice}}$ at the bottom. In general the water is pushed down at the top with a higher rate $v_{\text{sk}}$ than the ice gives in at the bottom with $v_{\text{ice}}$. The difference gives the squeeze flow, which has the parabolic Poiseuille profile.

The flow and the pressure are characterised by parameters $a$, $b$ and $c$. We here only need the pressure at the top and bottom of the layer, of which the dependence on the transverse $y$ direction is given by Eq. (47) as

$$p = \eta[-ay^2 - 2by + c], \tag{7}$$

with $\eta = 1.737 \cdot 10^{-3}$ Pas. The parameters of the side layer are indexed as $a_s, b_s, c_s$ with the coordinate $y$ replaced by $\zeta$ and $z$ by $\xi$. Those at the bottom are $a_b, b_b, c_b$ with coordinate $y$ replaced by $\xi$ and $z$ by $\zeta$. Thus we have to determine six parameters. Five of them follow from the boundary conditions on the pressure, the parameter $a_b$ determines the rate at which the water layer is squeezed out.

In previous calculations [6,9] the pressure was assumed to vanish at the edges of the skate in the upright case, in the idea that it should equal the outside air pressure, which is virtually zero as compared to the MPa pressures inside the layer. However, measurements [11] of the track left behind by the skate, indicate sharp walls in the ice caused by the edges. So at the edge the pressure must have been equal or larger than the hardness in order to give these plastic deformations. Therefore we expect the pressure in the basis water layer to exceed the hardness $p_{\text{h}}$.

On the other hand the formation of the side layer does not involve a (plastic) deformation and therefore the pressure in the side water layer will be below the hardness $p_h$. Continuity of the pressure then implies that the pressure at the edge, sandwiched by the two layers, will be equal to $p_h$. At the point where the side layer meets open air, we assume the pressure to vanish.

We first discuss the side layer where $y$ is replaced by $\zeta$. So the pressure Eq. (7) becomes

$$p_s(x,\zeta) = \eta[-a_s\zeta^2 - 2b_s\zeta + c_s]. \tag{8}$$

The first observation is that $a_s = 0$. It follows from the relation (52) between $a$ and the squeeze flow as derived in Appendix B. Both velocities vanish: $v_{sk} = 0$, because the skate does not move in the direction of the layer thickness and $v_{ice} = 0$, as the pressure in the side layer will not exceed the hardness. At the edge the variable $\zeta$ in the side surface assumes the value $\zeta_e$, implying the relation

$$\eta[-2b_s\zeta_e + c_s] = p_h. \tag{9}$$

For $\zeta = 0$ the pressure vanishes, which gives the second condition $c_s = 0$. Therefore $b_s$ has the value

$$b_s = -\frac{p_h}{2\eta\zeta_e}, \tag{10}$$

leading to the following expression for the pressure in the side water layer

$$p_s(x,\zeta) = p_h \frac{\zeta}{\zeta_e(x)}. \tag{11}$$

For the basis layer Eq. assumes the form

$$p_b(x,\xi) = \eta[-a_b\xi^2 - 2b_b\xi + c_b], \tag{12}$$

since now $\xi$ plays the role of the transverse coordinate $y$. For the basis layer the coordinate $\xi = 0$ at the edge. This gives the condition

$$p_h = \eta c_b. \tag{13}$$

At the other side of the basis layer, at the value $\xi = \xi_e$, we have again the pressure $p_h$, leading to the condition

$$p_h = \eta[-a_b\xi_e^2 - 2b_b\xi_e + c_b], \qquad \text{or} \qquad b_b = -a_b\xi_e/2. \tag{14}$$

Combining these relations we can write the pressure Eq. (7), in the basis layer as

$$p_b(x,\xi) = p_h + \eta a_b\xi(\xi_e(x) - \xi). \tag{15}$$

## 4   Frictional Melting

The principle of frictional melting is that the heat generated by friction melts a layer of the ice underneath the skate. A fraction of the heat leaks away into the skate and the other fraction melts the ice. If the temperature of the skate equals that of the ice, both fractions are 0.5. In [9] we have derived the equation (Eq. (16)) for the layer thickness $h(x)$

$$-\frac{\partial h}{\partial x} = \frac{k}{h} - \frac{1}{V}[v_{sk} - v_{ice}], \tag{16}$$

which describes the growth of the thickness $h$ downward along the skate. The first term on the right hand side gives the growth due to melting. This term is characterised by the small length $k$ ($\sim 10^{-11}$m)

$$k = \frac{\eta V}{2\rho L_h}, \tag{17}$$

with $\rho$ is the density of ice $\rho = 916.8$ kg/m$^3$ and $L_h$ the latent heat of melting $L_h = 0.334 \cdot 10^6$ J/kg. The factor 2 in the denominator follows from the fact that only half of the heat is available for melting. The second term in Eq. (16) accounts for the compression due to the rate $v_{\mathrm{sk}}$ at which the ice comes down and the rate $v_{\mathrm{ice}}$ at which ice gives in due to the pressure in the water layer.

The expression for $v_{\mathrm{sk}}$ follows from geometry of the skate. As mentioned $v_{\mathrm{sk}} = 0$ for the side layer. The downward velocity at the bottom layer follows from the curvature of the skate

$$v_{\mathrm{sk}} = V \frac{\partial \zeta(x)}{\partial x} = 2Vd\frac{x}{l^2} = V\frac{x}{R}. \tag{18}$$

### 4.1 Force and Friction at the basis

Using the imcompressibility of water we have derived in Appendix B the relation (52) between the squeeze flow and $a_b$. So Eq. (18) provides the fourth equation to find pressure $p$, the parameter $a_b$ and the two velocities $v_{\mathrm{sk}}$ and $v_{\mathrm{ice}}$. For the bottom layer the solution for $a_b$ reads

$$a_b = \frac{6Vx}{R[h_b^3 + 6\gamma\eta\xi(\xi_e - \xi)]}, \tag{19}$$

with the associated pressure given by

$$p_b(x,\xi) = p_{\mathrm{h}} + \frac{6\eta V x \xi(\xi_e(x)-\xi)}{R[h_b^3(x,\xi) + 6\gamma\eta\xi(\xi_e(x)-\xi)]}. \tag{20}$$

Hence we can make the layer equation (16) explicit by using Eq. (19) for $a_b$

$$-\frac{\partial h_b(x,\xi)}{\partial x} = \frac{k}{h_b(x,\xi)} - \frac{xh_b^3(x,\xi)}{R[h_b^3(x,\xi) + 6\gamma\eta\xi(\xi_e(x)-\xi)]}, \tag{21}$$

with $\xi_e(x)$ given by Eq. (5). The equation has to be integrated from $x = x_e(\xi)$, where $h_b(x_e(\xi),\xi) = 0$, downwards to $x = 0$. So for all values of $\xi$ within $0 \le \xi \le f$ we have to integrate Eq. (21) in order to find the layer thickness $h_b(x,\xi)$.

The bottom friction, due to the water layer is given by the integral

$$F_{\mathrm{bw}} = \eta \int_0^f d\xi \int_0^{x_e(\xi)} dx \frac{V}{h_b(x,\xi)}. \tag{22}$$

Apart from this friction we have also the ploughing friction as a result from making the indentation. It is given by the integral

$$F_{\mathrm{pl}} = \int_0^f d\xi \int_0^{x_e(\xi)} dx \, p_b(x,\xi)\frac{x}{R}. \tag{23}$$

The fraction $x/R$ gives the component of the force that has to be exerted in the forward direction.

The normal force exerted by the basis surface on the skate equals

$$F_{\mathrm{b}} = \int_0^f d\xi \int_0^{x_e(\xi)} dx \, p_b(x,\xi). \tag{24}$$

In these integrals the contact length enters as a trial parameter which has to be adjusted later such that the normal force matches the weight of the skater.

## 4.2 Force and Friction on the side

The layer equation for the water layer at the side simplifies since both $v_{sk} = 0$ and $v_{ice} = 0$. The former since the ice is not pushed down at the side and the latter since the pressure stays below $p_h$. So the layer equation becomes for side layer $h_s$

$$-\frac{\partial h_s}{\partial x} = \frac{k}{h_s}. \tag{25}$$

The integration starts from $x_e$ at the edge, with the solution

$$h_s(x) = [h_s^2(x_e) + 2k(x_e - x)]^{1/2}. \tag{26}$$

The initial condition $h_s(x_e)$ we find from the requirement that the outflow of the basic layer must match in inflow in the side layer at the edge of the skate where they meet. The amount of water leaving the bottom layer is at an edge point ($\xi = 0$)

$$\int_0^h dz v_y = -b_b h_b^3/6 = a_b \frac{\xi_e h_b^3}{12} = \frac{V \xi_e x_e}{2R}. \tag{27}$$

For the second equality we used Eq. (14) and for the third equality Eq. (19) with $\xi = 0$. Note that the thickness $h_b$ drops out of the relation. The same amount of water flows into the side layer at the same point at the edge ($\zeta = 0$)

$$\int_0^h dz v_y = -b_s h_s^3/6 = -\frac{p_h h_s^3}{12\eta \zeta_e(x)}, \tag{28}$$

where we used Eq. (10) for the second equality. Equating the results (27) and (28) gives an expression for $h_s$

$$h_s^3(x_e) = -\frac{6\eta V x_e \xi_e \zeta_e}{p_h R}. \tag{29}$$

Thus the calculations of the side layer can be carried out independently of the outcome of the basis layer, although they are connected by the requirement of continuity of the flow at the connecting edge of the skate.

We rewrite the expression a bit with the aid of Eq. (17) as

$$h_s^3(x_e) = -12\lambda \frac{k x_e \xi_e \zeta_e}{R}. \tag{30}$$

Using the values of $\rho$, $L_h$ and $p_h = 10$ MPa for ice, one has for the dimensionless combination $\lambda$

$$\lambda = \frac{\rho L_h}{p_h} \simeq 30. \tag{31}$$

We use the result (30) for the solution of the layer $h_s$. The integration of the thickness of the side water layer proceeds by fixing a value of $\zeta$ in the region $-d \leq \zeta \leq 0$ and starting the integration at $x_e(\zeta)$ on the edge. Writing $\xi_e$ also as function of the corresponding $\zeta$ via $\xi_e = -\zeta(f/d)$, the value of $h_s(\zeta)$ at the edge becomes the initial thickness $h_0(\zeta)$

$$h_0^3(\zeta) = 12\lambda \frac{k l f \zeta^2 \sqrt{1 + \zeta/d}}{dR}. \tag{32}$$

With this value of $h_0$ the solution for the side water layer reads more explicitly

$$h_s(x, \zeta) = [h_0(\zeta)^2 + 2k(x_e(\zeta) - x)]^{1/2}. \tag{33}$$

The friction is then, for a value of $\zeta$, given by

$$f_{\mathrm{fr}}(\zeta)) = \int_{-x_e(\zeta)}^{x_e(\zeta)} dx \frac{\eta V}{h_s(x,\zeta)}. \tag{34}$$

The integral is elementary and reads

$$f_{\mathrm{fr}}(\zeta)) = 2 p_{\mathrm{h}} \lambda \left( [h_0(\zeta)^2 + 4kx_e(\zeta)]^{1/2} - h_0(\zeta) \right). \tag{35}$$

The side friction is the integral

$$F_{\mathrm{sw}} = \int_{-d}^{0} d\zeta \, f_{\mathrm{fr}}(\zeta)), \tag{36}$$

which has to be performed numerically. The total friction is the sum of three contributions

$$F_{\mathrm{fr}} = F_{\mathrm{bw}} + F_{\mathrm{pl}} + F_{\mathrm{sw}}. \tag{37}$$

The pressure in the side layer is given by Eq. (11). The force on the side layer equals

$$F_{\mathrm{s}} = \int_{0}^{l} dx \int_{\zeta_e(x)}^{0} d\zeta \, p_s(x,\zeta) = \int_{0}^{l} dx \int_{\zeta_e(x)}^{0} d\zeta \, p_{\mathrm{h}} \frac{\zeta}{\zeta_x} = p_{\mathrm{h}} \frac{d\,l}{3}. \tag{38}$$

The larger the tilt, the larger $d$ and the larger the force on the side of the skate.

## 5 The Force Balance

The forces $F_{\mathrm{b}}$ and $F_{\mathrm{s}}$ of the ice on the skate, given by Eqns. (24) and (38), are normal to the surface. The components in the $z$ and $y$ direction are formed as the combinations

$$F_z = F_{\mathrm{b}} \cos \phi + F_{\mathrm{s}} \sin \phi, \qquad F_y = F_{\mathrm{b}} \sin \phi - F_{\mathrm{s}} \cos \phi. \tag{39}$$

The component in the $z$ direction balance the weight $Mg$ of the skater and the $y$ component balances the centrifugal force

$$F_z = Mg, \qquad F_y = \frac{MV^2}{R_{\mathrm{c}}}. \tag{40}$$

Thus we find the relation between the body inclination $\psi$ and the skate tilt $\phi$ as

$$\tan \psi = \frac{F_y}{F_z} = \frac{F_{\mathrm{b}} \sin \phi - F_{\mathrm{s}} \cos \phi}{F_{\mathrm{b}} \cos \phi + F_{\mathrm{s}} \sin \phi}. \tag{41}$$

Since both $F_{\mathrm{b}}$ and $F_{\mathrm{s}}$ are positive, $\psi < \phi$, as one observes from Fig. 1. An alternative form of Eq. (41) reads

$$\phi - \psi = \arctan(F_{\mathrm{s}}/F_{\mathrm{b}}), \tag{42}$$

which also follows directly from the balance of forces in the coordinate system of the skates. In order to get an impression of these tilt differences, we have plotted, in Fig. 4, $\phi - \psi$ as function of the velocity for a number of tilt angles $\phi$. The difference decreases slowly with the velocity and increases with the tilt angle. For a given tilt angle $\phi$ the radius of curvature has to chosen such that Eq. (2) is fulfilled. Velocities below 1 m/s make little sense in rounding a curve.

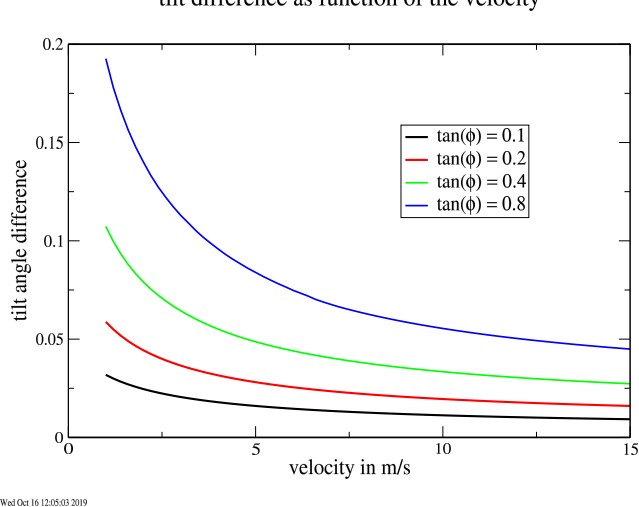

Figure 4: Difference $\phi - \psi$ as function of the velocity

## 6 Results

For the calculation of the friction one has to know the thickness of the water layers. That of the basis follows from the integration of Eq. (21) in the $x$ direction for each $\xi$ in the interval $0 \leq \xi \leq f$. If $f$ exceeds the width of the skate $w$ (as happens for very small tilt angles), the upper limit of the $\xi$ interval has to be replaced by $w$. The thickness of the side water layer is given by Eq. (33) for the values of $\zeta$ in the interval $-d \leq \zeta \leq 0$.

In general the side layer contributes a modest amount to the friction, only at large tilt angles and high velocities it starts to count. In Fig. 5 we have plotted the contributions of the friction of the water layers and the ploughing friction, together with the total friction as function of the tilt angle. The chosen velocity is $V = 10$ m/s. One observes that, while the water layer friction is rather insensitive, the ploughing friction increases as function of the tilt angle. As a result the friction in a curve is substantially larger friction than that of the upright skate. This happens already for small tilt angles. One should realise that a tilt of a few degrees easily occurs even for straightforward skating, rendering already some 20 % increase in the total friction.

In Fig. 6 we have plotted the friction as function of the velocity for various tilt angles, which can be translated with Eq. (3) in the radius $R_c$ of the curve. We see that the smaller the radius of the curve the larger the friction. Apart from a region of velocities smaller than walking speed, the increase sets in for all curvatures at higher speeds. Note that for $\tan(\phi) = 0.1$, which is a small tilt, the friction already rises from 1.2 at slow speeds to almost 2 for $V=10$ m/s. This is surprising since the concomitant equilibrium curvature of the stroke is $R_c = 100$ m for $V=10$ m/s, which is hardly distinguishable from a straight stroke.

By our choice of the boundary conditions on the pressure we have always a positive contribution to the force exerted by the ice on the side of the skate. This means that the resultant of the force on the basis and the side has a inclination $\psi$, (slightly) different from the tilt angle $\phi$ of the skate. The resultant points in the direction of the center of mass of the skater as seen from the skate on the ice.

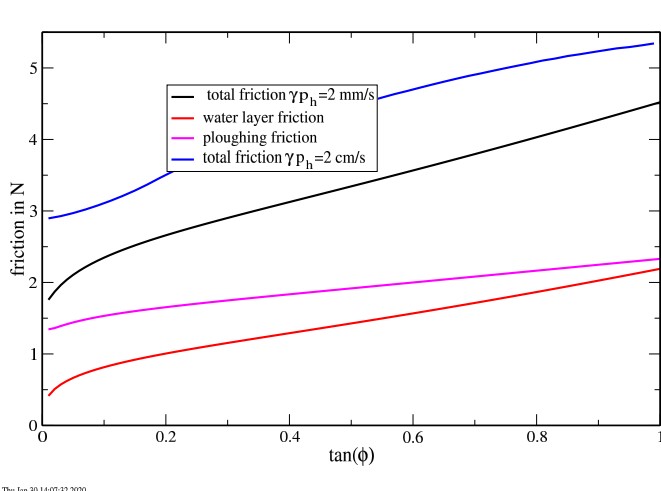

Figure 5: The friction contributions as function of the tilt angle

## 7  Conclusion

Using that skates trace a furrow in the ice by a plastic deformation, we have derived a set of equations from which the forces on the skate can be calculated by (numerical) integration of the layer equations for the basis and the side of the skate. We get the forces as function of the contact length. For a giving weight of the skate one must find the contact length by an iterative procedure from the normal force component on the ice matching with the weight of the skater.

The problem requires boundary conditions on the pressure in the water layer, resulting from melting due to the frictional heat. The plastic deformation implies that the pressure exceeds the hardness in the basis layer and stays below the hardness $p_h$ in the side layer. Continuity of the pressure then gives the value $p_h$ at the edge of the skate. As a result we find a small force on the side of the skate. In the equilibrium situation where the centrifugal force and the normal force are both balanced by the generated forces of the ice on the skate, we find the tilt angle $\psi$ of the body as function of the tilt angle $\phi$ of the skate.

Our calculation is a simplified version of the equations proposed for the upright position [6, 9] where heat generation and flows in the bulk are taken into account. In general these extensions are of minor influence except for more special circumstances as very low temperatures. We have focused on the influence of the tilt angle and find that the friction increases substantial with the tilt angle. In particular, for the unavoidable small tilts occurring in skating, the influence is large as shown in Fig. 5. Note that the friction increases with the velocity at fixed tilt angle, a feature which is stronger than the increase found in the upright skate. The ploughing force increases with the velocity but is tempered by following mechanism: due to the increasing pressure in the basis water layer, the skate is lifted (aqua planing) and therefore the skate makes a shorter contact with the ice, lowering the friction.

Comparing our approach with that the similar treatment by Lozowski et al. [7], we observe a number of differences.

- Here the inclination of the skate is related to the curvature of the track which gives a balance between the normal and centrifugal force. Hence we have a force on both the bottom layer *and* the side layer, whereas [7] has only a force on the bottom layer.

- We use a different rheology, Eq. (1). As we pointed out in the Introduction, this leads generally to lower frictions. In order to give an idea we show in Fig. (5) also the friction for $\gamma p_h = 2$cm/s, which is ten times the value used otherwise. As one sees this change makes an important difference.

- We employ different boundary conditions for the pressure in the bottom water layer. We set the pressure of the bottom water layer at the boundaries in the transverse direction equal to the hardness, since the skate deforms the ice in an inelastic (plastic) way and for plastic deformation one needs pressures higher than the hardness. This gives also a lower friction, but less than the change in $\gamma$. In appendix C we calculate the difference for the upright skate, which amounts an decrease of 0.05 N in the friction for a wide range of velocities.

- We have left out heat leak in the skates and ice due to unavoidable temperature gradients. One could add easily these terms to the layer equation, see [9], but as pointed out there, the change is not large.

The results of the calculation of [7] are closer to the measured friction in [10] than ours, which are too low. In general it is less worrisome to find a lower friction than a higher friction as compared to real skating occurs, since idealisations are made which favour gliding, such as perfectly smooth skates and ice and since parasite processes as heat loss to the ice are omitted. One could also interpret the difference as an indication that our chosen $\gamma$ is too low, but we feel that the used model is as yet too idealised to fit parameters like $\gamma p_h$ and $p_h$ to the experimental results.

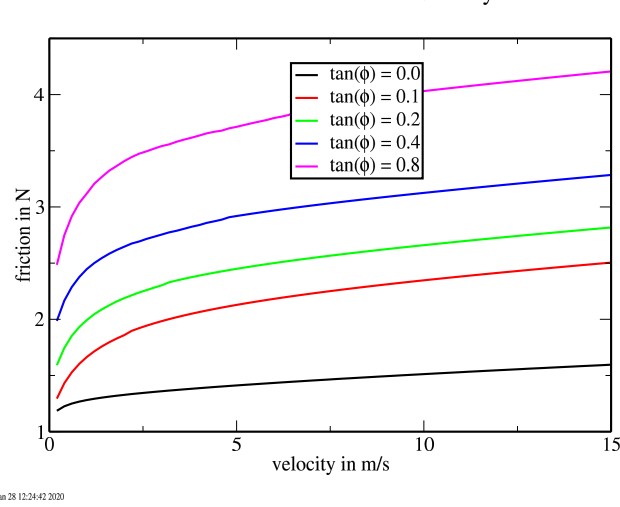

Figure 6: The friction as function of the velocity for various tilt angles

## A   The hydrodynamics of the water layer

We first give the shape of the flow field in a thin layer of thickness $h$ which exist in the region between $z = 0$ and $z = -h$. On top $z = 0$ the velocity in the $x$ direction equals $V$ of the skate

and at the bottom $z = -h$ it vanishes (stick boundary condition at the ice).

$$v_x = V\left(1 + \frac{z}{h}\right). \tag{43}$$

Due to the pressure water is also squeezed out sideways in the $y$ direction as a Poisseuille flow

$$v_y = -(ay + b)z(z + h). \tag{44}$$

At the top and bottom of the layer $v_y = 0$. The velocity in the $z$ direction has the profile

$$v_z = a\left(\frac{z^3}{3} + \frac{hz^2}{2} - \frac{h^3}{6}\right). \tag{45}$$

This component is dictated by the requirement of incompressible flow

$$\nabla \cdot \mathbf{v} = 0, \tag{46}$$

and the condition that $v_z = 0$ for $z = -h$.

The pressure which causes the flow in the $y$ direction has the form

$$p = \eta[-ay^2 - 2by + c + az(h + z)], \tag{47}$$

with $\eta = 1.737 \cdot 10^{-3}$ Pas, the viscosity of water. One verifies that the hydrodynamic equations

$$\nabla p = \eta \Delta \mathbf{v}, \tag{48}$$

are fulfilled. The constants $a$, $b$ and $c$ have to be obtained from the boundary conditions.

## B  Squeeze Flow

Consider a section $dx$ of the water layer at position $x$ with a thickness $h(x)$ and a width $w(x)$ in the transverse direction. The flow out of this volume is given by

$$\frac{d\mathcal{V}}{dt} = dx \int_{h(x)}^{0} dz[v_y(w(x)) - v_y(0)]. \tag{49}$$

Using Eq. (44) we find

$$\frac{d\mathcal{V}}{dt} = dx\, a\, w(x) h^3(x)/6. \tag{50}$$

We can also compute the outflow from the difference between the rate $v_{sk}$ at which the top of the volume comes down and the rate $v_{ice}$ at which the bottom of the volume goes down

$$\frac{d\mathcal{V}}{dt} = dx\, w(x)[v_{sk} - v_{ice}]. \tag{51}$$

The two expressions must agree since water is incompresible. Thus

$$[v_{sk} - v_{ice}] = a h^3(x)/6, \tag{52}$$

which relates $a$ to the squeeze flow.

## C The upright skate

In the upright position the skate touches the ice the the interval $-w/2 < y < w/2$, where $w = 1.1$mm is the width of the skate. We find for the pressure, at the top or bottom of the water layer, the expression

$$p(x,y) = p_h + \eta a(x)\left(\frac{w^2}{4} - y^2\right).$$
(53)

The constant $p_h$ raises the pressure at the edges equal to the hardness, which was missing in the previous treatment [9]. It was shown there that to a good approximation we may connect $v_{ice}$ with the average

$$v_{ice} = \gamma \int_{-w/2}^{w/2} \frac{dw}{w}(p(x,y) - p_h) = \gamma\eta a(x)w^2/6.$$
(54)

Combining this with the incompressibility of water, as expressed in Eq. (20), gives the relation for $a(x)$

$$a(x) = \frac{6xV}{R(h^3(x) + \gamma\eta w^2)}.$$
(55)

Then we can write the layer growth equation explicitly as

$$-\frac{dh(x)}{dx} = \frac{k}{h(x)} - \frac{xh^3(x)}{R(h^3(x) + \gamma\eta w^2)}$$
(56)

and the pressure expression as

$$p(x) = p_h + \frac{\eta Vxw^2}{R(h^3(x) + \gamma\eta w^2)}.$$
(57)

Integration of Eq. (56) yields the layer thickness $h(x)$ and Eq. (57) gives the pressure profile. As compared to the earlier calculation [9] we find a friction of 0.05 N lower over a wide range of velocities.

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
