# Peer review of "The friction of tilted skates on ice"

_SciPost Physics Core, doi:SciPost Phys. 8, 059 (2020)_

## Round 1 · Referee Report · Anonymous (Referee 1) · 2020-1-10

Report
This paper (I will refer to it as JVL) presents a dynamical model of ice friction which predicts ice friction as a function of skate blade tilt angle and skater velocity.
In its essential physics, this paper is virtually identical to the model presented in an earlier paper, published six years ago in one of the top journals in the field. Specifically, I refer to:
Lozowski, E.P., Szilder, K. and Maw, S, 2013: A model of ice friction for a speed skate blade. Sports Engineering. DOI 10.1007/s12283-013-0141-z.
I will call this earlier paper LSM. Both JVL and LSM predict ice friction for a tilted and an upright skate. The essential physics in JVL and LSM are similar. Both JVL and LSM predict that friction increases substantially with skate inclination angle. The main difference seems to be that LSM agrees very well with published measurements (see LSM Figs 12, 13, 14 and 15), while JVL states that his predicted friction “values are still about half those found in real skating experiments”.
In view of this comparison, I would suggest that the present author should first of all acknowledge the existence of LSM (at present it is not included in the bibliography) and point out the similarities between the two models. He should also identify in what ways he considers his model to be an improvement on LSM. And he needs to explain why, if it is a theoretical improvement, his model’s agreement with experiments is not as good as that found in LSM. If the present author makes these changes, I am prepared to recommend publication. Otherwise, I must recommend that the paper be rejected on the basis that it is almost identical to an existing paper, but it does not improve on the existing paper when the theoretical results are compared with experimental results.
In its essential physics, this paper is virtually identical to the model presented in an earlier paper, published six years ago in one of the top journals in the field. Specifically, I refer to:
Lozowski, E.P., Szilder, K. and Maw, S, 2013: A model of ice friction for a speed skate blade. Sports Engineering. DOI 10.1007/s12283-013-0141-z.
I will call this earlier paper LSM. Both JVL and LSM predict ice friction for a tilted and an upright skate. The essential physics in JVL and LSM are similar. Both JVL and LSM predict that friction increases substantially with skate inclination angle. The main difference seems to be that LSM agrees very well with published measurements (see LSM Figs 12, 13, 14 and 15), while JVL states that his predicted friction “values are still about half those found in real skating experiments”.
In view of this comparison, I would suggest that the present author should first of all acknowledge the existence of LSM (at present it is not included in the bibliography) and point out the similarities between the two models. He should also identify in what ways he considers his model to be an improvement on LSM. And he needs to explain why, if it is a theoretical improvement, his model’s agreement with experiments is not as good as that found in LSM. If the present author makes these changes, I am prepared to recommend publication. Otherwise, I must recommend that the paper be rejected on the basis that it is almost identical to an existing paper, but it does not improve on the existing paper when the theoretical results are compared with experimental results.

---

## Round 1 · Referee Report · Anonymous (Referee 2) · 2020-1-15

Strengths
Nice problem, which would be very original if the same authors had not already publish in SciPost in 2017 a very similar problem.
Weaknesses
The article has been written very rapidly without taking care of too many details. Furthermore it relies upon many equations of ref.6
Report
This article studies the friction on the ice skates as a function of the inclination angle of the skater in curved track. The main results is that because of the hydrodynamic pressure of the melted ice the angle of the skate is not the exactly the same than that of the skater body. The problem is nice but the article relies upon many results of ref.6 of the same author which makes the reading quite annoying. The author could have at least quoted the number of the equation of ref.6, which he is referring too. Indeed the article gives the impression of having been written very fast giving just a list of equations with very short comments. For example is never said explicitly that l<<R, the limit of \xi are not well defined, figure 3 is too schematic and very difficult to understand. The lis could be continued. Finally I do not understand how the length l disappear from the final results. This length should be related to the other parameter but I do not understand how.
As a conclusion the article is an application of the theory developed in ref 6 published in SciPost in 2017. Thus the novelty is very weak but as I said the problem is nice. Taking into account that the previous article has been published on the same journal I am not against in considering the present article for publication in SciPost. In any case it cannot be published in the present form and a substantial rewriting is necessary. The author has also to explain if there are new physics effects with respect to ref.6.
As a conclusion the article is an application of the theory developed in ref 6 published in SciPost in 2017. Thus the novelty is very weak but as I said the problem is nice. Taking into account that the previous article has been published on the same journal I am not against in considering the present article for publication in SciPost. In any case it cannot be published in the present form and a substantial rewriting is necessary. The author has also to explain if there are new physics effects with respect to ref.6.
Requested changes
Substantial rewriting

---

## Round 2 · Referee Report · Anonymous (Referee 2) · 2020-2-16

Strengths

It is an interesting problem of friction, whose solution is rather complex

Weaknesses

Originality. Indeed it is an extension (although interesting and difficult ) of a previous work published by the same author in this journal

Report

The author took seriously into account my previous remarks and now the article is rather clear and self contained.

---

## Round 2 · Referee Report · Anonymous (Referee 1) · 2020-2-24

Strengths

  1. The author has responded suitably to the comments of the second referee.

  2. It is now clearer what "improvements" have been made to the model, in terms of ice rheology and flow boundary conditions.

  3. An important missing paper has been added to the references.

Weaknesses

  1. A graph showing a direct comparison with experimental results, especially those of de Kooning et al. and with the theoretical results of Lozowski et al. would, in my opinion, be a benefit to the paper. However, I do not consider it to be essential to this paper, as I am confident that this comparison will eventually be done and published.

Report

The author has taken the comments of referee 2 seriously and made appropriate improvements to the paper. I now can recommend publication.

Requested changes

None required, but the author should consider the suggestion made in the section entitled "weaknesses".

---

## Round 2 · Author Response

Reply to the referees.

The first referee states that the paper is not very original and difficult
to read. Indeed the extension of the computation of the friction of an
upright skate to an inclined skate is not very original, but the technical
aspects of this calculation are far from trivial.
The paper was set up in a compact way since the author did not
want to repeat himself in the used material of the first publication.
In view of the criticism of the first referee the paper has been
rewritten paying attention to the detailed points raised by the first
referee. Concerning the role of the contact length,
it is not clear to the author what the referee means by the question
that the dependence on the contact length seems to be absent.
The contact length is THE important parameter in the calculation
determining the forces. Rather than plotting the friction as function
of the contact length, the contact length is determined by
the weight of the skater, which is the practical input parameter.
That requires an iterative procedure. This point has been stressed
stronger in the new version.

The second referee rightfully points to a very relevant paper
(of Lozowski et al.) which has been overlooked in the present study.
The author thanks the referee for this remark. The relation of the
present work to this earlier is worked out in detail in the new version.
The author disagrees with the second referee that the paper is
``almost identical to the paper mentioned''.
The difference with the papers of Lozowski et al. concerns the
rheology of ice with respect to external pressure and the boundary
conditions on the pressure in the layers of water between skate
and ice, in particular with respect to the forces on the side of the
blade. These differences are extensively elaborated in the new
version of the paper. That the outcome of the theory gives a lower
value of the friction (and therefore a larger difference with the
measurements) is in the opinion of the author not a drawback of the
new approach, given the uncertainty in the
hardness and response of ice and the idealisations of the model
(perfectly smooth skate and ice and omission of heatleaks in the ice).

---

## Round 2 · List of Changes

• In the Introduction the commentary on the rheology is made more explicit.
  • The justification of the used boundary conditions is given in the Introduction and the separate Section on the boundary conditions is rewritten such as to make it more explicit.
  • The role of the contact length is stressed and elaborated.
  • The discussion of the geometry (Fig. 3) is hopefully made more clear.
  • References to the previous paper are made more complete.
  • The missed publication of Lozowski et al. (new reference [6]) is extensively discussed both in the Introduction and the Conclusions, explaining the differences between this paper and the present paper.
  • Fig. 5 has been extended with one more curve and Fig. 6 has been corrected (horizontal axis).
  • Together these changes have extended the first version with two more pages.

---

## Editorial Decision

published